# Evaluation of Grand Ethiopian Renaissance Dam Lake Using Remote Sensing Data and GIS

**Asem Salama** [1,2,*] **, Mohamed ElGabry** [1,2] **, Gad El-Qady** [1] **and Hesham Hussein Moussa** [1,2]

[1] National Research Institute of Astronomy and Geophysics, Helwan 11421, Egypt
[2] African Disaster Mitigation Research Center (ADMIR), Helwan 11421, Egypt
* Correspondence: asem.mostafa@nriag.sci.eg; Tel.: +20-21224017195

**Abstract:** Ethiopia began constructing the Grand Ethiopian Renaissance Dam (GERD) in 2011 on the Blue Nile near the borders of Sudan for electricity production. The dam was constructed as a roller-compacted concrete (RCC) gravity-type dam, comprising two power stations, three spillways, and the Saddle Dam. The main dam is expected to be 145 m high and 1780 m long. After filling of the dam, the estimated volume of Nile water to be bounded is about 74 billion $m^3$. The first filling of the dam reservoir started in July 2020. It is crucial to monitor the newly impounded lake and its size for the water security balance for the Nile countries. We used remote sensing techniques and a geographic information system to analyze different satellite images, including multi-looking Sentinel-2, Landsat-9, and Sentinel-1 (SAR), to monitor the changes in the volume of water from 21 July 2020 to 28 August 2022. The volume of Nile water during and after the first, second, and third filling was estimated for the Grand Ethiopian Renaissance Dam (GERD) Reservoir Lake and compared for future hazards and environmental impacts. The proposed monitoring and early warning system of the Nile Basin lakes is essential to act as a confidence-building measure and provide an opportunity for cooperation between the Nile Basin countries.

**Keywords:** Grand Ethiopian Dam; GIS; the first, second and third storages; satellite data

## 1. Introduction

The construction of massive hydraulic infrastructures, such as big dams, has expanded to an unprecedented level around the world in the 20th century. With their influence on social and political relations, they are also shaped by political, social, and cultural conditions [1,2]. The downstream countries in the main world river system are generally opposed to the upstream project dams [3,4]. These dam projects cause many concerns in the downstream countries because of their possible social and environmental impacts, including droughts, water salinity, and water flow effects. In the Euphrates Basin, the downstream countries of Iraq and Syria were affected by four droughts in 2000, 2006, 2008, and 2009, which are a cascading effect of climate change and a large number of dams being constructed along the Euphrates River, which is known as the Southeastern Anatolia (GAP) Project [5,6]. The GAP project includes the construction of 22 dams and 19 hydraulic power plants for irrigation and the generation of electricity on the Euphrates and the Tigris rivers and their tributaries [2,5]. The Three Gorges Dam (TGD) was constructed in China in the Yangtze River, affecting the sediment discharge and regulation of the flow process in the downstream provinces, which resulted in severe scouring and changes in the hydrogeological regime [7]. Dam projects were established along the Mekong River from 1965 to 2019 in northeastern Thailand, China, Vietnam, Loas, and Cambodia for power electricity generation [8]. These dam projects have environmental, economic, river hydrogeology, biological, and sediment transfer effects in Myanmar, Laos, Thailand, China, Cambodia, and Vietnam [9].

In April 2011, Ethiopia started the construction of the Grand Ethiopian Renaissance Dam (GERD). Understanding the context of the dam and its position relative to other dams on the

Blue Nile is essential. The newly built dam is located downstream of the Tana Lake, a highland lake at an average altitude of 1800 m a.s.l., with a surface area of 3060 km$^2$ at an average lake level of 1786 m a.s.l. This lake has a maximum depth of 15 m [10]. Four major tributaries feed the Tana Lake sub-basin, the Gelgel Abay in the south, Rib and Gomera in the east, and Megech in the north (Figure 1). The GERD is a gravity roller-compacted concrete dam with a target height of 145 m and length of 1780 m. The dam's crest is supposed to be at a height of 655 m above sea level, with the prospective to impound a lake with a capacity of 74 billion m$^3$ [11]. About 116 km upstream of the GERD, the Rosaries Dam is located in Sudan, constructed in 1961 and heightened in 2013, with a current storage capacity of 7.4 billion m$^3$ (Nile Basin Atlas Program) [12]. About 100 km downstream of Rosaries, the Sennar Dam was constructed in 1926 with a capacity of about 390 million m$^3$ (Nile Basin Atlas Program) [12]. Further north in Sudan is the Meroe Dam, with an impoundment capacity of about 12.3 billion m$^3$. Further to the north in the very south of Egypt, the Aswan High Dam was constructed in 1970 and is considered to bee the last dam near the mouth of the Blue Nile. The total capacity of the Aswan High Dam is 164 billion m$^3$. It consists of dead storage of 31.6 billion m$^3$, active storage of 90 billion m$^3$ (BCM), and emergency storage for flood protection of 41 billion m$^3$ (Nile Basin Atlas Program) [12].

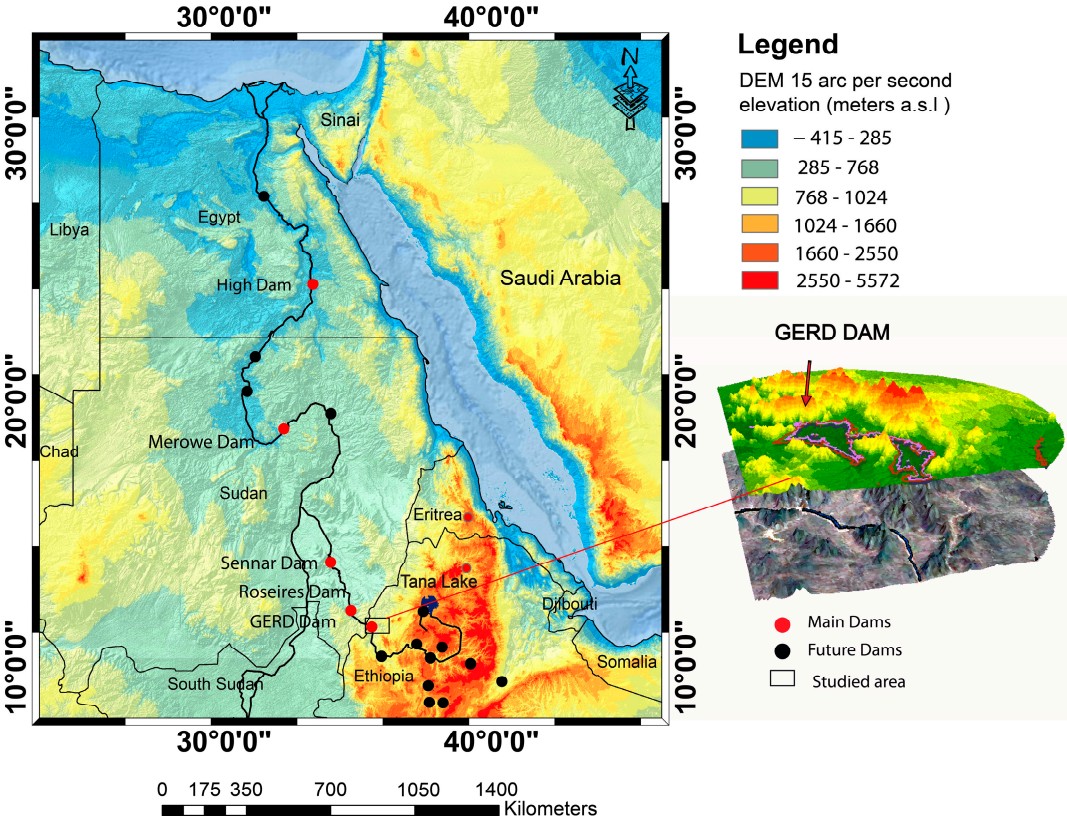

**Figure 1.** The location map of the studied area, which includes the location of the main artificial and proposed future dams in Africa. Data sourced from Wheeler et al. [13].

The Eastern Nile Basin is affected by historical complex hydropolitics over the use of the Nile water [14,15]. In the summer of 2020, the first phase of construction of the GERD was finished, and shortly after, the first filling of the GERD Lake started. During this season, the Sudanese dams, especially of Rosaries and Sennar, were confusedly operated due to a lack of prior information about the size and timing of the filling (reported by the Sudanese Minister of Irrigation Yasser Abbas on 26 August 2021 *Daily News* [16]. This may be due to the frozen agreement of the Eastern Nile Basin Initiative (NBI) activities [14,15]. This resulted in a shortage in freshwater during June and July, the filling months, in the capital Khartoum and many other cities after Sudanese water treatment stations went out

of service due to low river levels. Later, in the same season, after the end of filling, Sudan faced a vast flood as the level of the Nile reached 17.48 m on 27 August 2020 at Khartoum, which was considered the second-highest level after the 1912 flood according to Prime Minister Abdalla Hamdouk [17] (*Guardian Journal*; date 5 September 2020). Ninety-nine people were killed in this flood as mentioned by the state of emergency in Sudan. Ramadan et al. [18] referred to the negative impacts, including environmental, economic, and social problems, on Egyptian countries by applying different scenarios along 2, 3, and 6 years of the filling of the GERD under different flow conditions. Omran and Negm [19] considered the different filling scenarios and indicated that Egypt and Sudan would experience severe impacts during the filling phase of GERD in some scenarios.

Remote sensing has been used to estimate and monitor the volume of lakes worldwide in various case studies. The key parameters controlling the water quantity of small or large lakes are the area and top level [20–22]. The spatial and temporal changes in the volume of water bodies can be calculated by several methods depending on the availability of morphometric and areal data. Amitrano et al. [21] used DEM (9 to 15 m resolution obtained from SAR data) to estimate the depth. They analyzed both Sentinel-1 and COSMO-SkyMed imagery to obtain more accurate results to extract the boundary of the basin as the water level increased, reflected by increases in the contour, to estimate the reservoir surface volume and retained water volume of the reservoir in the Labaa Basin in Ghana region. Xiaoqi et al. [23] used STRM DEM of the above lake level to construct the relationship between the elevation and the area to estimate the volume of the Namsto Lake in China. Pipitone et al. [22] used both optical (Landsat 5 TM, Landsat 8 OLI-TIRS and ASTER images) and synthetic aperture radar (SAR) images to monitor the water surface and the level of the Castello Dam Reservoir. They defined the displacement using the global navigation satellite system (GNSS) to detect the relationship between the water level and dam deformation in Castello Dam on Magazzolo Reservoir in south Italy. Ahmed et al. [24] used the time series of Landsat images of 2001, 2011, and 2019 to extract the modified normalized difference water index and combined it with field observation water level data to calculate the lake volume from 2001 to 2019 in Deeper Beel, which is situated in the southwestern part of Guwahati, Assam in India. Jiang et al. [25] used the average annual coefficients of the VH backscatter for Sentinel-1A and the normalized difference water index (NDWI) of Sentinel-2 to map small water bodies in the mountain region in China for water-related environment monitoring and resource management. In the Nile Basin, Hossen et al. [26] built bathymetric and water capacity relationships based on Sentinel-3 optical and radar data for Aswan High Dam Lake, Egypt. Kansara et al. [27] used an analysis of multi-source satellite imagery and Sentinel-1 SAR imagery to display the number of classified water pixels in the GERD from early June 2017 to September 2020, indicating a contrasting trend in August and September 2020 for all upstream/downstream water bodies using a Google Earth Engine (GEE). Their results show that upstream of the dam rises steeply while it decreases downstream.

In the last 20 years, multispectral remote sensing and Sentinel (SAR-1) data have been widely used for surface water monitoring to overcome the limitations and lack of field observations for monitoring of the storage volume of water reservoirs [19,21,23,28]. The dynamic volume change in GERD Lake is essential for all Blue Nile countries, including Ethiopia, Sudan, and Egypt, to understand the balance of the water security

## 2. Methods and Materials

### 2.1. Depth Estimations

The depth of the GERD Lake was estimated using Shuttle Radar Topography Mission (SRTM) data, which map the topography of the Earth's surface using radar interferometry. The Shuttle Radar Topography Mission (SRTM) is an international project spearheaded by the National Geospatial-Intelligence Agency and NASA, whose objective is to obtain the most complete high-resolution digital topographic database of the Earth. We downloaded the SRTM 1 arc per second data courtesy of the U.S. Geological Survey from https://

earthexplorer.usgs.gov/ accessed on 21 June 2020. It was measured on 11 March 2000. It was used in the study of the GERD Lake to obtain the elevation difference obtained through interferometry, which was transformed into a 3D digital elevation model (DEM), which was used as the GERD Basin Reservoir depth before the filling process.

*2.2. Satellite Data Processing and Water Level Estimation*

In this study, we tracked changes in the water capacity level boundary for the GERD Lake using the multi-optical satellite data and Sentinel 1A (SAR). We acquired the multi optical Sentinel 2A and Landsat-8 with time series from 21 July 2020 to 3 July 2021 courtesy of the U.S. Geological Survey, https://ers.cr.usgs.gov/ website accessed on 21 June 2020 while the Landsat-9 and Sentinel-1 (SAR) with time series were from 16 July 2021 to 28 August 2022. The sentinel-1 (SAR) was obtained from Copernicus Open Access Hub https://scihub.copernicus.eu/ website accessed on 29 August 2022. The Sentinel-2 data was characterized by higher spatial and spectral resolutions in the near-infrared region. The Sentinel-2 sensor, the EO satellite of the Copernicus program, has 12 bands with spatial resolutions of 10 (four visible and near-infrared bands), 20 (six red-edge and shortwave infrared bands), and 60 m (three atmospheric correction bands) [29]. Recently, the Landsat-9 satellite was launched on 27 September 2021. It is similar to Landsat-8 and characterized by four visible spectral bands, one near-infrared spectral band, three shortwave-infrared spectral bands at a 30 m spatial resolution, plus one panchromatic band at a 15 m spatial resolution, and two thermal bands at a 100 m spatial resolution. The problem of dense cloud cover is encountered in some optical satellite imagery, which masks the lake in rainy seasons, especially in June and July each year. We used a filter to remove cloud pixels, using the threshold to identify the pixel range as cloud using ArcGIS 10.8 software [30]. We found incomplete filter-out cloud in some multi-optical satellite images. We instead used Sentinel-1 SAR to obtain the water level boundary, especially in the cloud periods, which mask the GERD Lake boundaries. SAR sentinel-1 (Synthetic Aperture Radar (S-1 SAR) data are insensitive to cloud. However, Sentinel-1 SAR data are characterized by speckle noise and have some difficulties in detecting the water surface of water bodies. This can be solved by applying several techniques such as aggregation of the brightness pixels, which was proposed by Pipitone et al. [22].

The analysis scheme used to estimate the water volume in the GERD Lake is summarized in Figure 2 for optical multispectral and SAR data analysis, which was applied in this study. We used ArcMap 10.8 [30] for multioptical satellites (i.e., LandSat-8 and -9 and Sentinel-2) to separate the shape of the GERD Lake using the normalized difference water index (NDWI) as it enhances the presence of water bodies, a method introduced by Mcfeeders [31]. NDWI uses reflected near-infrared radiation and visible green light to enhance the presence of water bodies such as lakes and rivers. This method is characterized by its ability to eliminate the presence of soil and terrestrial vegetation features. The equation depends on the use of bands with a relatively high reflectance of the water green band (band-3) and one with low or no reflectance near-infrared (NIR) (i.e., band-8 in the case of multispectral Sentinel-2 and band-5 in the case of Landsat-8 and -9) as follows:

$$NDWI = \frac{Band\ 3 - Band\ 8\ or\ 5\ (NIR)}{Band\ 3 + Band\ 8\ or\ 5\ (NIR)} \tag{1}$$

The preprocessing of the workflow of Sentinel SAR-1 was applied by the Sentinel Application Platform (SNAP) [32], an open-source software version of 8.0.9 (http://step.esa.int/main/toolboxes/snap/ accessed on 1 October 2020), as follows: (a) a subset tool was used to delineate the area of the study. (b) The orbit file was applied, which allows updating of the orbit state vectors for each SAR scene, providing accurate satellite position and velocity information. (c) The thermal to noise removal algorithm was used to remove and reduce noise effects in the inter-sub-swath texture and normalize the backscatter for scenes in multi-swath acquisition modes. (d) Calibration equation was used to convert the image intensity values to sigma nought values in which the digital pixel was converted to

radiometrically calibrated SAR backscatter concerning the nominally horizontal plane of Sentinel-1 GRD. (e) Terrain corrections were used to compensate for some distortions related to the side-looking geometry to be close to the real world. (f) We used coregistration with an average stack of two time series images per month to obtain a single image. We applied coregistration instead of a speckle filter to remove noise without affecting the resolution of the optical image of Sentinel SAR-1, which may result from temporal decorrelation effects. The final step was to convert it to linear transformations and apply the band math equation depending on the image histogram. The water lake was delineated using this equation in which the thresholding values range between −1, which refers to land, and +1, which refers to water bodies.

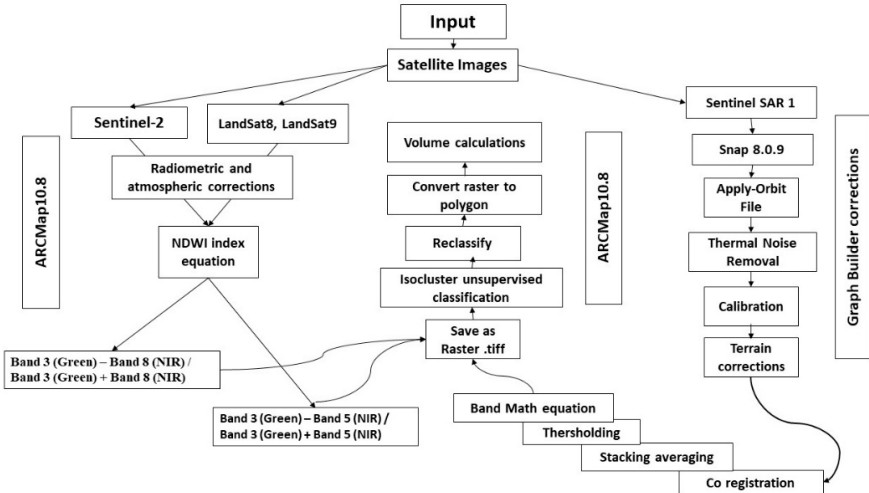

**Figure 2.** Workflow explaining the steps of satellite image analysis carried out in this study to calculate the volume of the GERD Lake.

### 2.3. Water Level Validations

The water levels were collected for Nasser, Tana, and GERD lakes in Egypt and Ethiopia, respectively. The in situ water level data was recorded by the gauging station at Nasser Lake and obtained from the Nile Research Institute (NRI) database. The other water level data was collected at virtual stations from a satellite altimetry set obtained from the "Global Reservoirs and Lakes Monitor (G-REALM) project" of the U.S. Foreign Agricultural Service [33] and the level contour was extracted by optical multispectral satellite data in this study. Then, we calculated the average water level uncertainty as shown in Table 1.

**Table 1.** Water level for the Aswan, Tana, and GERD lakes.

| Name Lake | Water Level from In Situ Station | Water Level from Virtual Stations Obtained from Satellite Altimeter Data from G-REALM) Project "m" | Location of Virtual Water Level Station | | Water Level Extracted by Sentinel-2 Boundary "m" in This Study | Differences "m" |
|---|---|---|---|---|---|---|
| | | | Long. | Lat. | | |
| Nasser lake, Aswan Egypt | 180.5 | 181.93 | 32.57 | 22.8 | 181 | 1.43 |
| Tana Lake, Ethiopian | - | 1789.31 | 37.3 | 12.0 | 1787.81 | 1.6 |
| GERD Lake, Ethiopian | - | 581.38 | 10.579 | 10.552 | 580 | 1.4 |
| Average calculated water level uncertainty | | | | | | ±1.45 |

## 3. Results and Volume Calculation

The required parameters needed to compute the volume of the GERD Lake were as follows: (a) the input surface (i.e., 3D depth of the lake), which was established from the digital elevation model. (b) The second parameter required is the "Z" value, which was defined as the plane surface height of the water level top boundary in which the lake polygons were extracted from an optical satellite image or Sentinel 1A–SAR. The volume

equation was calculated using the ARCGIS10.8 volume tool, which was dependent on the empirical formula of the volume. The volume equation is as follows:

The volume of water bodies = average depth (d) of the Lake X Area of the lake (A)    (2)

The computation of the DEM raster surface was evaluated using the extent of the center point of each cell as opposed to the extent of the entire cell area. The resulting analysis will decrease the data area of the raster by half a cell relative to the data area displayed for the raster according to the manual of ARCGIS10.8.

The average volume uncertainty was calculated with the average uncertainty in volume (Figures 3 and 4) depending on the water level uncertainty ± 1.45 calculated in the previous section. The lakes' polygons' boundaries were extracted from a multi-optical satellite image and Sentinel SAR-1 to reflect the water area storage morphology in the GERD Lake (Figure 3). A chart of the average volume for the GERD Lake with the time series obtained every month from 21 July 2020 to 28 August 2022 is shown in Figure 4.

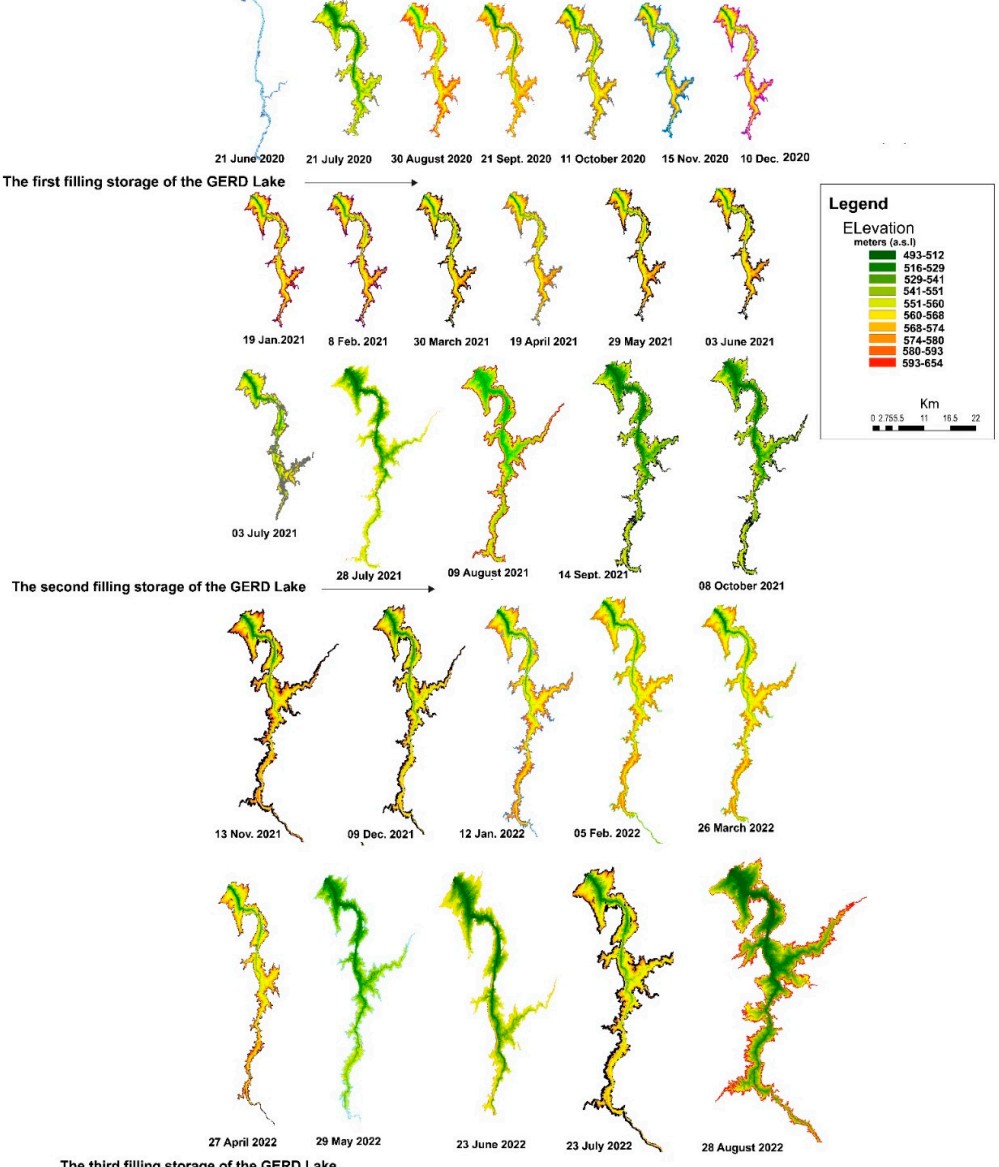

**Figure 3.** Digital elevation model (DEM) extracted by the GERD Lake polygons with time series from satellite images.

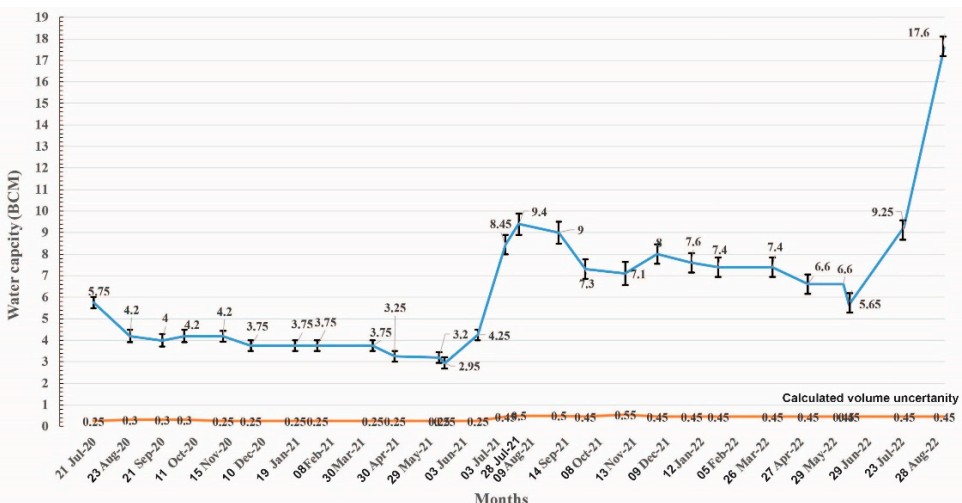

**Figure 4.** Chart showing the water volume capacity of the GERD Lake with the calculated volume uncertainty values in orange color from 21 July 2020 to 28 August 2022.

The volume of the GERD Lake in the first storage reached its maximum level, which appeared in the satellite images taken on 21 July 2020, with an area of 250.16 km$^2$ and a volume of 5.75 $\pm$ 0.25 billion m$^3$ (Figure 4). Although, there was a receding of the water in the GERD Lake in the next three months in August, September, and October in the year 2020, with an average water volume of 4.2 $\pm$ 0.3 billion m$^3$ (Figure 4). From November 2020 to 30 March 2021, the second receding of water storage in the GERD Lake reached an average volume of 3.75 $\pm$ 0.3 billion m$^3$, calculated from the satellite images. On 28 July 2021, the GERD Lake showed an increase in the polygon area extracted from the Sentinel SAR-1 satellite image of 316.54 km$^2$ and an average volume of 8.45 $\pm$ 0.45 billion m$^3$ (Figures 3 and 4). The average volume of storage of the GERD Lake increased in August and September 2021, with a maximum of 9.4 $\pm$ 0.5 billion m$^3$ during August 2021 after the second storage was carried out. Receding of the water of the lake was observed during October and November 2021 to an average volume of 7.3 $\pm$ 0.45 billion m$^3$ (Figure 4), with a slight increase in December 2021 to 8.0 $\pm$ 0.45 billion m$^3$. From January to 29 May 2022, the capacity of the reservoir lake decreased to 5.56 $\pm$ 0.45. Then, the third filling storage was reached by 23 July 2022, with an increase in the total capacity of 9.25 $\pm$ 0.25 billion m$^3$ and a significant large capacity of 17.4 $\pm$ 0.45 billion m$^3$ was reached on 28 August 2022 (Figures 3 and 4).

The Ethiopian government carried out the first storage in July 2020 while July 2021 and July 2022 represent the second and third storage stages. During the storage stages and closing of the GERD Dam gates, the GERD Lake was charged by rainfall and Tana Lake, which is considered the major source of the Blue Nile [10].

The water level was observed from satellite images to be on the lower limit of the saddle dam in the third filling on 28 August 2022. This saddle dam was built with a 5-km-long concrete face rockfall and 50 m high to maintain the required water surface elevation and depth at a relatively flat dam site. The saddle dam increases the natural features from 600 to 646 m asl, increasing the reservoir water level to the design level [34]. An emergency gated 300-m-wide spillway is located between the main dam and the saddle dam. The spillway, at a crest elevation of 624.9 m, is to be used for extreme flood conditions, releasing through a gully into the river downstream of the dam.

## 4. Discussion

The application of remote sensing and GIS to monitor GERD Lake volume changes provides critical information about the GERD Reservoir Lake water level and storage capacity. This will be very important for downstream countries in the case of a limitation or lack of information resulting from a stumble in negotiations between Ethiopia and

the downstream countries Egypt and Sudan. Water safety is essential for both upstream and downstream countries. One of the most controversial debuts in GERD negotiations is the number of years for the initial reservoir filling, as a shorter filling time requires greater flow reduction and a higher investment return from the dam. A longer filling time requires lower flow reduction and lower investment return from the dam [34]. The water level shown by satellite data in this study was 600 ± 1.45 a.s.l on August 28 August 2022 in the lower level of the saddle dam. This level corresponds to 24.3% of the full storage capacity of 74 billion cubic meters. It was considered as more than the minimum reservoir fill rates, which is beneficial for hydroelectric generation without having an effect on stream flow into Egypt and Sudan as stated by Keith et al. [35]. King and Block [36] refer to the 25% filling policy, which can reduce the average downstream flow by more than 10 BCM per year. Hegay et al. [37] proposed numerous actions and mitigation strategies that could secure Egypt's water demands by minimizing the effects of the GERD project. These strategies should include the present-day operation of the AHD hydropower plant to mitigate imminent water shortages in combination with the increase in groundwater withdrawal as a backstop choice to quickly sustain the water demand. Water conservation strategies should additionally be integrated, mainly inside the agriculture sectors, by switching the countrywide production to crops that require less water.

Previous studies have investigated the possible future multi-environmental and hazard impacts on downstream countries. Wheeler et al. [38] described a post-filling period that includes severe multi-year droughts after filling of the dam with the uncertainty of the exact start and end time, which will require careful coordination to minimize possible harmful impacts on downstream countries. Donia and Negm [39] modeled three scenarios of the storage capacity of the GERD Lake. The storage capacity of the three models was estimated assuming 18 billion $m^3$ for the initial design storage capacity and 35 and 74 billion $m^3$ for the middle and final storage. Their results from scenario-3 of the full filling of GERD Lake in 5 years show a negative impact on agriculture due to the loss of silt, which is a result of restricting the water flowing to the Aswan High Dam in Egypt. Abulnaga's [40] study refers to scooping out accumulated mud and silts through dredging and the construction of onshore sediment ponds that are used for agricultural purposes due to the construction of the dam in Ethiopia. From an engineering point of view, EL Askary et al. [41] showed a deformation pattern associated with different sections of the GERD Lake and Saddle Dam (main dam and embankment dam) using 109 descending mode scenes from Sentinel-1 SAR imagery from December 2016 to July 2021. This may result in a dam failure flood, which will have harmful impacts in Sudan and Egypt.

In summary, the environmental impacts and other socio-political considerations of GERD extend across a diverse spectrum of issues from population growth, economic development, and water rights to sedimentation and/or changing flood regimes and the shock of climate change. It is necessary to examine the complex social and environmental values of water resources and the policies governing the use of water resources. A water cooperation policy is the best choice for the cooperative Nile basin initiative to overcome any debate on the remnant years of fillings [42]. Informal diplomacy has been successfully used to manage transboundary waters in a similar case in the Mekong River Dam [43]. The waterscape of the Mekong Dam issues has been extended to security actors that are not water experts within domestic politics. For this, the analysis could be extended to examine in more detail the knowledge channels within multiple tracks of diplomacy and how harms and inequalities are understood, beyond mere metrics of economic impacts and water quantities. This method of informal diplomacy can help change the frozen negotiation situations between Ethiopia and Egypt. Thus, understanding water diplomacy requires scrutiny of how power, knowledge, and the political economy of river basin development intersect.

## 5. Conclusions

The combination of open-source satellite optical and radar images with DEM provided a robust tool to estimate the water volume in the artificial GERD Lake during the initial

phases of filling. The water level measured from satellite data refers to the consequent increase in the stored water volume of the GERD Reservoir Lake. Three stored water stages of the initial filling were considered for the lake, corresponding to volumes of $5.75 \pm 0.25$, $9.4 \pm 0.5$, and $17.4 \pm 0.45$ billion m$^3$ during 21 July 2020, 28 July 2021, and 28 August 2022, respectively.

Data collected from open sources combined with technical knowledge could provide very useful information that can be used to monitor the filling process and support informal diplomacy with transparent and trustful independent information that could possibly lead to a future agreement between all Nile basin countries. The authors believe that this work is a milestone in building a scientific initiative to utilize open-source data for the benefit of the community and to build a common agreement on the importance of investment in knowledge for sustaining water resources and their management. Further work is needed to extend this work to better understand the impact of the current filling process and its impact on the ecosystem and boost the knowledge and data exchange between riparian countries for integrated management plans for the Nile. An integrated database that combines ground- and satellite-based observations could utilize modern scientific techniques to integrate the dam's operation process and mitigate natural disasters and climate change's impact on the sustainable development in Nile Basin countries. Such an initiative could work as a confidence-building measure between Nile Basin countries and provide leveraging for science diplomacy to bridge cooperation and integration in an era of divergence and competition.

**Author Contributions:** Conceptualization, M.E. and G.E.-Q.; Data curation, A.S. and M.E.; Formal analysis, A.S.; Funding acquisition, M.E. and G.E.-Q.; Investigation, M.E.; Methodology, A.S., M.E., A.S. and H.H.M.; Writing—original draft, A.S. and M.E.; Writing—review and editing, H.H.M. and G.E.-Q. All authors have read and agreed to the published version of the manuscript.

**Funding:** This research received no external funding.

**Institutional Review Board Statement:** Not applicable.

**Informed Consent Statement:** Not applicable.

**Data Availability Statement:** Data and related results of this article are presented in the paper and can be asked from the first and second authors.

**Acknowledgments:** The authors are grateful to the Copernicus Sentinel-hub (https://scihub.copernicus.eu/) (accessed on 9 August 2022) and USGS https://earthexplorer.usgs.gov/ (accessed on 21 June 2020) for the availability of satellite data sources used in this research work. We would like to thank the National research institute of Astronomy and geophysics (NRIAG) for their financial and logistic support.

**Conflicts of Interest:** The authors declare no conflict of interest.

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
