# Peer review of "Evaluation of Grand Ethiopian Renaissance Dam Lake Using Remote Sensing Data and GIS"

_water, doi:10.3390/w14193033_

Round 1

Reviewer 1 Report (Previous Reviewer 1)

Authors have used remote sensing techniques and a Geo-graphic information system to monitor the changes in the volume of water in Grand Ethiopian Renaissance dam reservoir. It is essential to act as a confidence-building measure and provide an opportunity for cooperation between the Nile basin countries.

 For there are some grammar mistakes, this manuscript needs to be edited by a native Englishman. Therefore, I suggest authors submit it to the journal once again after solving the following problems.

 1.         Line 64, to unify name for the full text, "Grand Dam" in Figure 1 left image was suggested to be modified as "GERD Dam"

2.         Line 29, please check and rewrite the sentence.

3.         Line 91, 2.1 “Literature survey” can be moved to the section 1 “Introduction”.

4.         Line 144, please check and rewrite the sentence.  

5.         Lines 214-216, please check and rewrite the sentence.

6.         Line 271, section 4 “Discussion” should be rewritten to discuss (1) the advantages of water monitoring by Remote sensing and GIS, (2) the minimum and maximum water volumes stored in Grand Ethiopian Renaissance Dam Lake to ensure the water safety in Ethiopian and other countries, such as Sudan and Egypt. Furthermore, (3) future hazards and environmental impact can also be discussed here.  

Please present them clearly by sequence numbers throughout the manuscript.

7.         Line 306, section 5 “Conclusions” should be rewritten to show your main findings, such as (1) the principles of lake water change, (2) dangerous water stored level and (3) the assumption of how to use your findings in the flood early warning system.  

Author Response

Response to Reviewer 1 Comments

Thank you for comments and your notes. we have put the comment in our consideration in introduction, result and conclusions sections according to the reply to comments as follow:

Point 1: Authors have used remote sensing techniques and a Geo-graphic information system to monitor the changes in the volume of water in Grand Ethiopian Renaissance dam reservoir. It is essential to act as a confidence-building measure and provide an opportunity for cooperation between the Nile basin countries.

For there are some grammar mistakes, this manuscript needs to be edited by a native Englishman. Therefore, I suggest authors submit it to the journal once again after solving the following problems.

Response 1: We have been checken the Grammer and spelling mistakes. In addation to we will addreses this to jouranl to check the language by native english person

Point 2: Line 64, to unify name for the full text, "Grand Dam" in Figure 1 left image was suggested to be modified as "GERD Dam"

Response 2: It has been updated in line 86 at the revised manuscript version

Point 3: Line 29, please check and rewrite the sentence.

Response 3: we have rewrite in lines 31 to 34 at the revised manuscript version

Point 4: Line 91, 2.1 “Literature survey” can be moved to the section 1 “Introduction”.

Response 4:  we have moved to lines 92 to 126 at the revised manuscript version

Point 5: Line 144, please check and rewrite the sentence.  

Response 5:  we rewrote thes secnetence in updated lines 142 to 148 at the revised manuscript.

Point 6: Lines 214-216, please check and rewrite the sentence.

Response 6:  we rewrote thes secnetence in updated lines 215 to 225 at the revised manuscript.

Point 7: Line 271, section 4 “Discussion” should be rewritten to discuss (1) the advantages of water monitoring by Remote sensing and GIS, (2) the minimum and maximum water volumes stored in Grand Ethiopian Renaissance Dam Lake to ensure the water safety in Ethiopian and other countries, such as Sudan and Egypt. Furthermore, (3) future hazards and environmental impact can also be discussed here. 

Please present them clearly by sequence numbers throughout the manuscript.

Response 7:  we rewrote the Disscussion section in the updated lines 280 to 335 at the revised manuscript

Point 8: Line 306, section 5 “Conclusions” should be rewritten to show your main findings, such as (1) the principles of lake water change, (2) dangerous water stored level and (3) the assumption of how to use your findings in the flood early warning system

Response 8:  we rewrote the conclusions in this arrange form in the updated lines 338 to 349 at the revised manscrpit

Response to Reviewer 1 Comments

Thank you for comments and your notes. we have put the comment in our consideration in introduction, result and conclusions sections according to the reply to comments as follow:

Point 1: Authors have used remote sensing techniques and a Geo-graphic information system to monitor the changes in the volume of water in Grand Ethiopian Renaissance dam reservoir. It is essential to act as a confidence-building measure and provide an opportunity for cooperation between the Nile basin countries.

For there are some grammar mistakes, this manuscript needs to be edited by a native Englishman. Therefore, I suggest authors submit it to the journal once again after solving the following problems.

Response 1: We have been checken the Grammer and spelling mistakes. In addation to we will addreses this to jouranl to check the language by native english person

Point 2: Line 64, to unify name for the full text, "Grand Dam" in Figure 1 left image was suggested to be modified as "GERD Dam"

Response 2: It has been updated in line 86 at the revised manuscript version

Point 3: Line 29, please check and rewrite the sentence.

Response 3: we have rewrite in lines 31 to 34 at the revised manuscript version

Point 4: Line 91, 2.1 “Literature survey” can be moved to the section 1 “Introduction”.

Response 4:  we have moved to lines 92 to 126 at the revised manuscript version

Point 5: Line 144, please check and rewrite the sentence.  

Response 5:  we rewrote thes secnetence in updated lines 142 to 148 at the revised manuscript.

Point 6: Lines 214-216, please check and rewrite the sentence.

Response 6:  we rewrote thes secnetence in updated lines 215 to 225 at the revised manuscript.

Point 7: Line 271, section 4 “Discussion” should be rewritten to discuss (1) the advantages of water monitoring by Remote sensing and GIS, (2) the minimum and maximum water volumes stored in Grand Ethiopian Renaissance Dam Lake to ensure the water safety in Ethiopian and other countries, such as Sudan and Egypt. Furthermore, (3) future hazards and environmental impact can also be discussed here. 

Please present them clearly by sequence numbers throughout the manuscript.

Response 7:  we rewrote the Disscussion section in the updated lines 280 to 335 at the revised manuscript

Point 8: Line 306, section 5 “Conclusions” should be rewritten to show your main findings, such as (1) the principles of lake water change, (2) dangerous water stored level and (3) the assumption of how to use your findings in the flood early warning system

Response 8:  we rewrote the conclusions in this arrange form in the updated lines 338 to 349 at the revised manscrpit

Response to Reviewer 1 Comments

Thank you for comments and your notes. we have put the comment in our consideration in introduction, result and conclusions sections according to the reply to comments as follow:

Point 1: Authors have used remote sensing techniques and a Geo-graphic information system to monitor the changes in the volume of water in Grand Ethiopian Renaissance dam reservoir. It is essential to act as a confidence-building measure and provide an opportunity for cooperation between the Nile basin countries.

For there are some grammar mistakes, this manuscript needs to be edited by a native Englishman. Therefore, I suggest authors submit it to the journal once again after solving the following problems.

Response 1: We have been checken the Grammer and spelling mistakes. In addation to we will addreses this to jouranl to check the language by native english person

Point 2: Line 64, to unify name for the full text, "Grand Dam" in Figure 1 left image was suggested to be modified as "GERD Dam"

Response 2: It has been updated in line 86 at the revised manuscript version

Point 3: Line 29, please check and rewrite the sentence.

Response 3: we have rewrite in lines 31 to 34 at the revised manuscript version

Point 4: Line 91, 2.1 “Literature survey” can be moved to the section 1 “Introduction”.

Response 4:  we have moved to lines 92 to 126 at the revised manuscript version

Point 5: Line 144, please check and rewrite the sentence.  

Response 5:  we rewrote thes secnetence in updated lines 142 to 148 at the revised manuscript.

Point 6: Lines 214-216, please check and rewrite the sentence.

Response 6:  we rewrote thes secnetence in updated lines 215 to 225 at the revised manuscript.

Point 7: Line 271, section 4 “Discussion” should be rewritten to discuss (1) the advantages of water monitoring by Remote sensing and GIS, (2) the minimum and maximum water volumes stored in Grand Ethiopian Renaissance Dam Lake to ensure the water safety in Ethiopian and other countries, such as Sudan and Egypt. Furthermore, (3) future hazards and environmental impact can also be discussed here. 

Please present them clearly by sequence numbers throughout the manuscript.

Response 7:  we rewrote the Disscussion section in the updated lines 280 to 335 at the revised manuscript

Point 8: Line 306, section 5 “Conclusions” should be rewritten to show your main findings, such as (1) the principles of lake water change, (2) dangerous water stored level and (3) the assumption of how to use your findings in the flood early warning system

Response 8:  we rewrote the conclusions in this arrange form in the updated lines 338 to 349 at the revised manscrpit

Reviewer 2 Report (New Reviewer)

It was a pleasure to read the paper on the impacts of the GERD. I have a few comments to strengthen the paper:

-          The lines numbers must be in English, not in Arabic

-          Why in page one certain sentences are underlines?

-          Lines 31-34 in page 1, should also include references to the work of Ahmet Conker on the hydropolitics of the Euphrates River:

Conker, Ahmet, et al. "Hydropolitics and issue-linkage along the Orontes River Basin: An analysis of the Lebanon–Syria and Syria–Turkey hydropolitical relations." International Environmental Agreements: Politics, Law and Economics 20.1 (2020): 103-121.

Conker, Ahmet. "BÜYÜK HÝDROLÝK YAPILAR-SOÐUK SAVAÞ-SINIRAÞAN SU SORUNLARI ÝLÝÞKÝSÝ VE SOÐUK SAVAÞ'IN BIRAKTIÐI MÝRAS." Alternatif Politika 11.2 (2019): 319-340.

Conker, Ahmet, and et al. "Hydraulic mission at home, hydraulic mission abroad? Examining Turkey’s regional ‘pax-aquarum’and its limits." Sustainability 11.1 (2019): 228.

Conker, Ahmet. (2022). Small is beautiful but not trendy: Understanding the allure of big hydraulic works in the Euphrates-Tigris and Nile waterscapes. Mediterranean Politics, 27(3), 297-320.

-          On the issue of the dams on the Mekong River, please also read the work of Filippo Menga and of Naho Mirumachi

-          In page 3, you say “that there was no prior arrangement with downstream countries”. Nevertheless, you should mention that there have been discussions and attempts to find agreements, which did not lead to an agreed arrangement. See and include the following work:

Cascão, A. E., & Nicol, A. (2016). GERD: new norms of cooperation in the Nile Basin?. Water International41(4), 550-573.

Yihdego, Zeray, Alistair Rieu-Clarke, and Ana Elisa Cascão. "How has the Grand Ethiopian Renaissance Dam changed the legal, political, economic and scientific dynamics in the Nile Basin?." Water International 41.4 (2016): 503-511.

Hussein, H., & Grandi, M. (2017). Dynamic political contexts and power asymmetries: The cases of the Blue Nile and the Yarmouk Rivers. International environmental agreements: Politics, law and economics17(6), 795-814.

Hussein, H., & Grandi, M. (2015). Contexts matter: a hydropolitical analysis of Blue Nile and Yarmouk River basins. Social water studies in the Arab Region159, 159-176.

These articles needs to be used also in the introduction to show the context of the Nile hydropolitics relations, which at the moment is quite shallow.

-          Page 3 would also benefit from more review of the latest work of Kevin Wheeler and of Ana Elisa Cascao. On the political side and on the lack of cooperation, see also the latest piece of Kevin Wheeler, who discusses why it is difficult to have a shared vision and agreement on the Nile: Wheeler, Kevin G., et al. "Water research and nationalism in the post-truth era." Water International 46.7-8 (2021): 1216-1223.

Moreover, please justify better the methods used, and link the literature review (with the suggestions I provided above) with the discussion and concluding remarks, as at the moment it feels a bit missing this link.

Author Response

Response to Reviewer 2 Comments

Thank you for your valuable comments and your notes. we have put the comments in our consideration with improvement the introduction, results, Discussions and conclusions sections according to reply to the comments as follows:

Point 1: It was a pleasure to read the paper on the impacts of GERD. I have a few comments to strengthen the paper:

The lines numbers must be in English, not in Arabic

Response 1: we use the English lines numbers

Point 2: Why on page one certain sentences are underlines?

Response 2: we removed the underline

Point 3 Lines 31-34 on page 1, should also include references to the work of Ahmet Conker on the hydro politics of the Euphrates River:

Conker, Ahmet, et al. "Hydropolitics and issue-linkage along the Orontes River Basin: An analysis of the Lebanon–Syria and Syria–Turkey hydro political relations." International Environmental Agreements: Politics, Law and Economics 20.1 (2020): 103-121.

Conker, Ahmet. "BÜYÜK HÝDROLÝK YAPILAR-SOÐUK SAVAÞ-SINIRAÞAN SU SORUNLARI ÝLÝÞKÝSÝ VE SOÐUK SAVAÞ'IN BIRAKTIÐI MÝRAS." Alternatif Politika 11.2 (2019): 319-340.

Conker, Ahmet, et al. "Hydraulic mission at home, hydraulic mission abroad? Examining Turkey’s regional ‘pax-aquarium and its limits." Sustainability 11.1 (2019): 228.

Conker, Ahmet. (2022). Small is beautiful but not trendy: Understanding the allure of big hydraulic works in the Euphrates-Tigris and Nile waterscapes. Mediterranean Politics, 27(3), 297-320.

Response 3: we add a section from 25 to 32 in the revised manuscript

   Point 4  On the issue of the dams on the Mekong River, please also read the work of Filippo Menga and Naho Mirumachi

Response 4 we add the works of Zeitoun et al. 2017 on lines 61 to 63 in the revised manuscript. We add sentences about the work of Noho Mirumachi in the Discussions section to find a solution to diplomacy that succeeded in the Mekong river 328 to 335.

Point 5   In page 3, you say “that there was no prior arrangement with downstream countries”. Nevertheless, you should mention that there have been discussions and attempts to find agreements, which did not lead to an agreed arrangement. See and include the following work:

Cascão, A. E., & Nicol, A. (2016). GERD: new norms of cooperation in the Nile Basin. Water International41(4), 550-573.

Yihdego, Zeray, Alistair Rieu-Clarke, and Ana Elisa Cascão. "How has the Grand Ethiopian Renaissance Dam changed the legal, political, economic and scientific dynamics in the Nile Basin?." Water International 41.4 (2016): 503-511.

Hussein, H., & Grandi, M. (2017). Dynamic political contexts and power asymmetries: The cases of the Blue Nile and the Yarmouk Rivers. International environmental agreements: Politics, law and economics17(6), 795-814.

Hussein, H., & Grandi, M. (2015). Contexts matter: a hydro political analysis of Blue Nile and Yarmouk River basins. Social water studies in the Arab Region159, 159-176.

These articles need to be used also in the introduction to show the context of the Nile hydropolitics relations, which at the moment is quite shallow.

Response 5 We added the lines 67 to 68 at the section of introduction and 72-73

   Point 6   Page 3 would also benefit from more review of the latest work of Kevin Wheeler and of Ana Elisa Cascao. On the political side and the lack of cooperation, see also the latest piece of Kevin Wheeler, who discusses why it is difficult to have a shared vision and agreement on the Nile: Wheeler, Kevin G., et al. "Water research and nationalism in the post-truth era." Water International 46.7-8 (2021): 1216-1223.

Response 6 we added sections 324 to 326

Point 7   Moreover, please justify better the methods used, and link the literature review (with the suggestions I provided above) with the discussion and concluding remarks, as at the moment it feels a bit missing this link.

Response 7 we have taken in consideration these points in the revised manuscript

Round 2

Reviewer 1 Report (Previous Reviewer 1)

Section 5 “Conclusions” is too short to make your findings clear. 

Please expand it to at least a half page. 

Author Response

Response to Reviewer 1 Comments

Thank you for your comments and your notes.

Point 1: (x) Moderate English changes required

Response 1: We  have proofread the manuscript with corrections of English and misspellings

Point 2: Are the conclusions supported by the results? can be improved

Section 5 “Conclusions” is too short to make your findings clear.

Please expand it to at least a half page.

Response 2:  We have updated the conclusions section in the revised manuscript on lines 336 to 351.

Reviewer 2 Report (New Reviewer)

Looks better now.

Author Response

Response to Reviewer 2 Comments

Thank you for your valuable comments and your notes.

Point 1: Moderate English changes required

Response 1: We  have proofread the manuscript with corrections of English and misspellings

Point 2: Looks better now.

Response 2: Thanks.

This manuscript is a resubmission of an earlier submission. The following is a list of the peer review reports and author responses from that submission.

Round 1

Reviewer 1 Report

Authors have used remote sensing techniques and a Geo-graphic information system to monitor the changes in the water volume of Grand Ethiopian Renaissance dam reservoir. Which is of essential to act as a confidence-building measure and provide an opportunity for cooperation between the Nile basin countries.

Some suggestions are given as follows. 

1. Line 19. Abbreviations such as "GERD" should be given full name at the fist time used.

2. Line 43. Abbreviations such as "BCM" should be given full name at the fist time used.

3. Lines 45-46. To unify name for the full text, "Grand Dam" in Figure 1 was sugguested to be modified as "GERD Dam" 

4. Line 101. Problems existing in dam monitoring are suggested to be proposed here to explain why you do this work. 

5. Lines 187-195. Equations on water volume calculation and their accuracy evaluation are suggested to be added in section 3.

Reviewer 2 Report

Dear Authors,

as you can see below, I have many doubts regarding your work. I hope that my comments will help you in better structuring your future research.

Introduction

I suggest providing a more general context before discussing the case study. Despite the importance of the GERD dam, it is worth providing some comments on how it compares to similar projects, to show why it is worth studying it. 

Methodology

The review of the literature is incomplete and the cited documents are not very related to the study. I suggest revising the study rationale and then looking at what past studies can be cited.

Results

The results are affected by some assumptions (e.g., constant depth of the lake). I suggest expanding this part to avoid confusion.

Comparing satellite-derived results with field evidence is needed to confirm your approach. Please provide more evidence on the study validation.

Why do you describe the case study and the region in the Results section (lines 223-228)? This can generate a lot of confusion.

Discussion

Please use this part to compare your results with similar studies, not only related to the GERD, but also to reservoirs experiencing similar problems. 

Conclusions

Please rewrite this section summarizing the main outcomes of the study.

Overall structure

The manuscript should be restructured, to guarantee a clear picture and to allow readers to follow the main points of the study. In the present version, the literature review is confused with the study case presentation, and the results are not well discussed.
